# Modeling Within-Host Dynamics of SARS-CoV-2 Infection: A Case Study in Ferrets

**DOI:** 10.3390/v13081635

**Published:** 2021-08-18

**Authors:** Naveen K. Vaidya, Angelica Bloomquist, Alan S. Perelson

**Affiliations:** 1Department of Mathematics and Statistics, San Diego State University, San Diego, CA 92182, USA; abloomquist@sdsu.edu; 2Computational Science Research Center, San Diego State University, San Diego, CA 92182, USA; 3Viral Information Institute, San Diego State University, San Diego, CA 92182, USA; 4Los Alamos National Laboratory, Theoretical Biology and Biophysics Group, Los Alamos, NM 87545, USA; asp@lanl.gov

**Keywords:** COVID-19, ferrets, SARS-CoV-2, viral dynamics, within-host modeling

## Abstract

The pre-clinical development of antiviral agents involves experimental trials in animals and ferrets as an animal model for the study of SARS-CoV-2. Here, we used mathematical models and experimental data to characterize the within-host infection dynamics of SARS-CoV-2 in ferrets. We also performed a global sensitivity analysis of model parameters impacting the characteristics of the viral infection. We provide estimates of the viral dynamic parameters in ferrets, such as the infection rate, the virus production rate, the infectious virus proportion, the infected cell death rate, the virus clearance rate, as well as other related characteristics, including the basic reproduction number, pre-peak infectious viral growth rate, post-peak infectious viral decay rate, pre-peak infectious viral doubling time, post-peak infectious virus half-life, and the target cell loss in the respiratory tract. These parameters and indices are not significantly different between animals infected with viral strains isolated from the environment and isolated from human hosts, indicating a potential for transmission from fomites. While the infection period in ferrets is relatively short, the similarity observed between our results and previous results in humans supports that ferrets can be an appropriate animal model for SARS-CoV-2 dynamics-related studies, and our estimates provide helpful information for such studies.

## 1. Introduction

Severe acute respiratory syndrome coronavirus 2 (SARS-CoV-2) is a novel coronavirus that causes the infectious disease COVID-19 [1,2,3,4]. The ongoing global pandemic of COVID-19 has spread to almost all countries globally, with more than 178 million confirmed cases and more than 3.8 million deaths as of 21 June 2021 [5]. While tremendous efforts have been put into the control of COVID-19 outbreaks, these have mainly been non-pharmaceutical. A number of different vaccines have been approved. However, the disease is still spreading because of vaccine hesitancy, the lack of availability of vaccines in most parts of the world, and the difficulties of getting vaccines into arms when it is available. The emergence of SARS-CoV-2 variants amplifies already existing substantial threats to global public health [6,7].

The viral kinetics of SARS-CoV-2 during infection within an individual is poorly understood. The viral load data from human infection is rarely available during the early infection period [8,9,10,11,12]. In the past, animal models have successfully substituted for human hosts to study the within-host dynamics of many viruses, such as HIV, HCV, and HBV [13,14,15,16,17]. In particular, respiratory viruses that infect humans have been widely studied using ferrets [18,19,20,21,22,23,24,25,26], and recent studies have shown that SARS-CoV-2 can successfully transmit and replicate in ferrets [1,27,28,29,30,31]. As in previous studies [32,33,34,35,36,37,38,39], mathematical models fit to data from animals can be a powerful tool to characterize within-host viral kinetics of SARS-CoV-2. Such information is helpful to design, evaluate, and identify antiviral agents that can control SARS-CoV-2 infection within COVID-19 patients [1].

There are limited studies on within-host SARS-CoV-2 viral dynamics [8,9,40,41,42,43,44,45,46,47]. While previous studies have provided important insights into the viral dynamics [40,42,43,44], immune responses [8,9,45], and potential antiviral therapy [41,48] within human hosts [8,9,40,41,42,44,45] and macaques [40,43], many aspects of within-host dynamics still remain uncertain. Furthermore, most of the data used in these studies were obtained post-symptom onset, and none of these models dealt with the dynamics within the ferret, which has been considered one of the important animal models for experimental studies of SARS-CoV-2 [1,27,28,29,30,31].

In this study, we used mathematical models to characterize SARS-CoV-2 infection dynamics in two groups of ferrets [1]: one infected with SARS-CoV-2/F13/environment/2020/Wuhan isolated from an environmental sample collected in the Huanan Seafood Market in Wuhan (F13-E) and another isolated from a person infected with SARS-CoV-2/CTan/human/2020/Wuhan (CTan-H). Specifically, we used the data from SARS-CoV-2-infected ferrets [1] in our models to estimate viral kinetic parameters, such as the infection rate, the viral clearance rate, the infectious virus proportion, the infected cell death rate, and the viral production rate. In addition, we further used our model to calculate the basic reproduction number, pre-peak infectious viral growth rate, post-peak infectious viral decay rate, pre-peak infectious viral doubling time, post-peak infectious virus half-life, and the total cell loss in the respiratory tract. Our results provide evidence-based quantitative insights into within-host viral dynamics in animal models that can significantly benefit the control of SARS-CoV-2 infection in humans.

## 2. Materials and Methods

### 2.1. Experimental Data

The data used in this study were obtained from digitizing a published experimental infection of ferrets by SARS-CoV-2 [1]. As mentioned above, two different SARS-CoV-2 viruses were considered: F13-E (isolated from an environmental sample) and CTan-H (isolated from a human patient). Six ferrets (three animals in each group) were inoculated intranasally with $10^{5}$ plaque-forming unit (PFU) of F13-E virus (animal: F13-E-1, F13-E-2, and F13-E-3) and CTan-H virus (animal: CTan-H-1, CTan-H-2, and CTan-H-3) in a volume of 1 mL. Viral RNA copies and viral titer (infectious virus) were recorded in nasal washes collected on days 2, 4, 6, 8, and 10 post infection (p.i.) from each animal. For our modeling, the data lying below the lower limit of detection (i.e., 3 $log_{10}$ viral RNA copies per mL and 1 $log_{10}$ PFU per mL) were taken as half of the limit of detection. 

### 2.2. Mathematical Model

To model the data containing viral RNA copies and viral titer (infectious virus), we used three viral dynamic models (Model 1: basic viral dynamics; Model 2: viral dynamics with eclipse phase; and Model 3: viral dynamics with immune response), similar to the ones for influenza [36,49], but we considered two types of virus population: infectious ($V_{i}$) and non-infectious ($V_{n}$). As SARS-CoV-2 was found to primarily replicate in the upper respiratory tract (nasal turbinate, soft palate, tonsils) of ferrets [1], we considered cells in the upper respiratory tract as targets of SARS-CoV-2 infection. Infectious SARS-CoV-2, $V_{i}$, infects target cells, T, at rate $\beta TV_{i}$, where β is a rate constant. Infected cells, I, are assumed to die at per capita rate δ and produce new free virus particles at rate p per cell. A portion, α, of the newly produced free viruses are assumed to be infectious, and the remaining $\left(1-\alpha \right)$ free viruses are non-infectious. The parameter c represents the clearance rate of both infectious and non-infectious viruses.

The system of equations in Model 1 is
$$\begin{array}{l} \frac{dT}{dt}=-\beta TV_{i},\;\;\;\;\;\;T\left(0\right)\;=\;T_{0},\\ \frac{dI}{dt}=\beta TV_{i}-\delta I,\;\;\;\;\;\;I\left(0\right)\;=\;I_{0},\;\\ \frac{dV_{i}}{dt}=\alpha pI-cV_{i},\;\;\;\;\;\;\;\;V_{i}\left(0\right)\;=\;V_{i0},\\ \frac{dV_{n}}{dt}=\;\left(1-\alpha \right)pI-cV_{n},\;\;\;\;\;\;\;V_{n}\left(0\right)\;=\;V_{n0}. \end{array}$$

In Model 2, we consider two classes of infected cells: one in the eclipse phase, $I_{e}$, which do not produce virus yet, and another in the productive phase, $I_{p}$, which are actively producing virus. The newly infected cells remain in the eclipse phase for an average duration of 1/k. With these mechanisms, the system of equations in Model 2 becomes
(1)$$\begin{array}{l} \frac{dT}{dt}=-\beta TV_{i},\;\;\;\;\;\;T\left(0\right)\;=\;T_{0},\\ \frac{dI_{e}}{dt}=\beta TV_{i}-kI_{e},\;\;\;\;\;\;I_{e}\left(0\right)\;=\;I_{e0},\\ \frac{dI_{p}}{dt}=kI_{e}-\delta I_{p},\;\;\;\;\;\;I_{p}\left(0\right)\;=\;I_{p0},\;\\ \frac{dV_{i}}{dt}=\alpha pI_{p}-cV_{i},\;\;\;\;\;\;\;\;V_{i}\left(0\right)\;=\;V_{i0},\\ \frac{dV_{n}}{dt}=\;\left(1-\alpha \right)pI_{p}-cV_{n},\;\;\;\;\;\;\;V_{n}\left(0\right)\;=\;V_{n0}. \end{array}$$

To develop Model 3, we extend Model 1 by incorporating an immune response. There are many potential immune responses, and incorporating all of the effects due to immune responses requires highly complex models. However, as a representative of immune response-models, we consider a simple approach because of limited data. SARS-CoV-2-specific adaptive immune response has not been properly documented in ferrets during the infection period (i.e., 10 days). Antibodies against SARS-CoV-2 in these ferrets have been measured, but only at day 13 and day 20 post infection [1]. Therefore, we consider the nonspecific innate immune response, which provides the first-line defense [50]. Specifically, we include type I Interferon (mainly IFN-α/β), which induces the expression of many IFN-stimulated antiviral proteins in the neighboring cells, making them refractory to infection [50]. We follow the modeling idea used by Pawelek et al. [36] to model such immune effect. We represent the IFN level by F and assume IFN makes uninfected cells refractory to infection with rate σ. The IFN level grows in proportion to the infected cells at rate g and decays at the per capita rate of ω. For simplicity, we assume the cells in the refractory state do not revert to the susceptible state in this relatively short infection period.

The system of equations representing Model 3 is
(2)$$\begin{array}{l} \frac{dT}{dt}=-\beta TV_{i}-\sigma TF,\;\;\;\;\;\;T\left(0\right)\;=\;T_{0},\\ \frac{dI}{dt}=\beta TV_{i}-\delta I,\;\;\;\;\;\;I\left(0\right)\;=\;I_{0},\;\\ \frac{dV_{i}}{dt}=\alpha pI-cV_{i},\;\;\;\;\;\;\;\;V_{i}\left(0\right)\;=\;V_{i0},\\ \frac{dV_{n}}{dt}=\;\left(1-\alpha \right)pI-cV_{n},\;\;\;\;\;\;\;V_{n}\left(0\right)\;=\;V_{n0},\;\\ \frac{dF}{dt}=\;gI-\omega F,\;\;\;\;\;\;\;F\left(0\right)\;=\;F_{0}. \end{array}$$

### 2.3. Parameters and Data Fitting

Cells in the upper respiratory tract are the main target for SARS-CoV-2 infection [1]. Based on the surface area per epithelial cell and the total area of epithelial cells lining the nasal turbinate of the human upper respiratory tract, Baccam et al. [49] calculated the total cells in the upper respiratory tract of humans to be $4\times 10^{8}$ cells. For simplicity, we scaled this value based on the ratio of the average weight of a human (75 kg) to the average weight of a ferret (2 kg) and thus assume there are $10^{7}$ epithelial cells in ferret upper respiratory tract. Only a fraction of these cells expresses the ACE2 receptor and enzyme TMPRSS2 needed to cleave the spike protein so that the virus can fuse to the cell membrane and successfully enter and infect a cell. Estimates of this fraction range from 1% [51] to 20% [52]. We will assume 10% and take $T_{0}=10^{6}$ cells. However, we note that the number of initial target cells chosen, *T*_0_, does not affect the results of this study because redefining $T\to T/T_{0},\;I\to I/T_{0},\;I_{e}\to I_{e}/T_{0},\;I_{p}\to I_{p}/T_{0},$ and $p\to pT_{0}$ will keep the systems unaltered, implying that the only parameter affected by a different choice of *T_0_* is *p*. Based on a previous study [41], we set g=1 per day and ω=0.4 per day [41].

As infection was established with a virus inoculum, we set $I_{0}=I_{e0}=I_{p0}=0$. To estimate the initial viral load inoculated, we obtained the regression using viral load and corresponding virus titer from all animals at all time points. Then, using initial virus titer of $10^{5}$ PFU in the regression formula, we obtained 9 log_10_ viral RNA copies, which we set to be the initial virus inoculated. The remaining parameters ($V_{i0},\beta ,\;\delta ,\alpha ,p,c$), ($V_{i0},\beta ,\;\delta ,\alpha ,p,c,k$), and ($V_{i0},\beta ,\;\delta ,\alpha ,p,c,\sigma$) in Model 1, 2, and 3, respectively, were estimated by fitting the models to experimental data from each ferret individually. We solved the model equations numerically using the built-in solver ode45 in MATLAB 2020a (The MathWorks, Inc.). The ode45 is based on an explicit Runge–Kutta (4,5) formula, the Dormand–Prince pair. The predicted log_10_ values of the total viral load ($V_{i}+V_{n}$) and the infectious virus concentration ($V_{i}$) were fitted to the corresponding log-transformed data via a nonlinear least square regression method in which the sum of the squared residuals (*SSR*), i.e., the difference between the model predictions and the corresponding experimental values, was minimized. We used the following formula to calculate *SSR*:(3)$$SSR=\sum_{t_{d}=1}^{N_{d}}[{\left\{\log_{10}V\left(2t_{d}\right)\;-\;\log_{10}\bar{V}\left(2t_{d}\right)\right\}}^{2}\;+\;{\left\{\log_{10}V_{i}\left(2t_{d}\right)\;-\;\log_{10}{\bar{V}}_{i}\left(2t_{d}\right)\right\}}^{2}],$$
where $V\left(t_{d}\right)\;=\;V_{i}\left(t_{d}\right)\;+\;V_{n}\left(t_{d}\right)$ represents the viral load at time $t_{d}$ predicted by the model, $V_{i}\left(t_{d}\right)$ represents the infectious virus concentration at time $t_{d}$ predicted by the model, and $\bar{V}\left(t_{d}\right)$ and ${\bar{V}}_{i}\left(t_{d}\right)$ are the values of viral RNA copies and the viral titer, respectively, in the experimental measurement. $N_{d}$ = 5 is the total number of time points at which measurements were taken, and the sum is taken over the set of measurement time points. In this study, a total of 10 data points, including two different types (viral load and infectious virus) was available for each fitting.

We also performed the global minimum search by providing various initial guesses distributed uniformly across the reasonable parameter limits. To overcome any possible stiffness of the model equations, we performed data fitting using ode15s, which is an ODE solver designed for stiff equations, and we did not find any difference in our fitting results. For each best-fit parameter, we also provide 95% confidence intervals (CI), which were computed from 250 replicates, by bootstrapping the residuals [53,54]. Since parameter values were constrained to be positive, potentially causing their distributions to be skewed to the right [49], we computed the geometric mean to report the average parameter values.

We also performed an analysis to determine whether the model parameters we expect to estimate are uniquely identifiable for the available viral load and infectious virus data. We note that while parameters may be correlated for the basic TIV (Target cell, Infected cell, and Virus particles) model [55], our model is a generalization that includes infectious virus, and we fitted to both the total viral load as in the TIV model as well as to the measured infectious titer. Thus, we used more data than in the TIV model. To address the identifiability issue, we generated the new data at the time of sampling from the model simulations with estimated known parameters. Then, we fitted the generated data with our fitting routine using different initial guesses as in our global minimization calculations. We found that the parameters, including both p and β, used to generate simulated data were recovered by our fitting procedure. We also repeated the process by adding 5%, 10%, 15%, and 20% random errors to the generated data. In this case, we obtained estimated values of the parameters approximately close to what we used for data generations but with slight variations, as expected. In addition, to quantify the correlation of parameters fitted using our data sets with two different natures (viral load and infectious virus), we also computed Pearson correlation coefficients for each model and each animal using the parameters obtained from replicates by bootstrapping the residuals. Our results show that the parameters estimated in our case are not correlated (Appendix A). Therefore, we do not expect identifiability issues in our fitting, while we acknowledge some uncertainty due to limitations on the models and data.

### 2.4. Sensitivity Analysis

To study how the model dynamics are affected by uncertainty present in model parameters, we considered a widely varying parameter space for each model. With these parameter spaces, we performed a global sensitivity analysis using Latin Hypercube sampling for each of the three models.

## 3. Results

### 3.1. Data Fitting to Three Viral Dynamics Models and Comparison

For each ferret, we fitted each of the three models to the experimental data. In the fitting of Model 1, we estimated the initial infectious virus concentration ($V_{i0}$), the infection rate constant (β), the death rate of infected cell (δ), the proportion of infectious virus among newly produced virus (α), the virus production rate (p), and the virus clearance rate (c). In Models 2 and 3, we estimated the additional parameters k (the rate of transfer from eclipse phase to infectious phase) and σ (the IFN-induced antiviral efficacy), respectively, along with those parameters estimated in Model 1. The best estimates of these parameters, along with 95% confidence intervals and geometric mean across ferrets, are given in Table 1 (Model 1), Table 2 (Model 2), and Table 3 (Model 3). With these parameters, the model predictions agree well with the experimental data on both viral RNA copies and viral titer (Figure 1—Model 1, Figure 2—Model 2, Figure 3—Model 3).

To examine which model can better fit, we computed the Sum of the Squared Residuals (SSR) and the Akaike Information Criterion (AIC) values. As discussed in the method section, SSR indicates how close the model prediction is to the data, and AIC allows one to compare models given the goodness of fit for a given model, number of data points, and number of parameters. Our model comparison shows that the basic model is approximately as good as the other two extended models (Model 1: 4.24 (SSR), 61.36 (AIC); Model 2: 4.14 (SSR), 151.16 (AIC); Model 3: 3.88 (SSR), 150.52 (AIC)), and thus the additional parameters k and σ, introduced in Models 2 and 3, respectively, did not improve the data fitting. Since the models provide similar dynamics, and the basic model has the best (lowest) AIC value, we present most of the results to follow based on Model 1. However, we observed that Models 2 and 3 also provide a similar conclusion about the viral dynamic properties.

The model fits revealed a trend in SARS-CoV-2 dynamics within a ferret in which the viral load initially decayed for about 17 h and then increased until it reached a peak around 5 days post infection. However, the concentration of infectious virus increased monotonically until it reached a peak. After the peak, both viral load and infectious virus decreased to a level below the limit of detection at around 8 days post infection. At the peak, the viral load reached about 8 log_10_ RNA copies per mL, among which ~4 log_10_ (~0.01%) are infectious viruses. The general pattern observed in SARS-CoV-2 dynamics is similar to that for influenza virus [36,49].

### 3.2. Important Viral Dynamics Parameters

The average estimate for the infection rate, β, was $6.93\times 10^{-5}$ per infectious virus per day. As indicated by our estimate of $\alpha =1.75\times 10^{-4}$, only about 0.02% of the total newly produced virus were infectious, and most of the viral particles were non-infectious. This low estimate of the newly produced infectious virus proportion is consistent with other data [56]. We also obtained that the infection in ferret was initiated with about 5 infectious viruses per mL on average (i.e., $V_{i0}\;\sim \;5$/mL), while the experiment was initiated with the intranasal inoculation of 10^5^ PFU of stock viruses in 1 mL. Therefore, only a very few infectious viruses may be needed at the infection site (upper respiratory tract) to initiate and establish SARS-CoV-2 infection. 

The infected cell death rate, δ, and virus clearance rate, c, estimated by our model were 5.20 per day and 6.10 per day, respectively. These estimates indicate that infected cells turned over in about 5 h on average, and the free virus gets cleared from the body in about 4 h on average. Using Model 2, we estimated k (the rate of cell transfer from the eclipse phase to the productively infected phase) to be 7.88 per day, which implies that the infected cells started producing virus about 3 h after they were infected. Similarly, we used Model 3 to estimate the rate constant for cells becoming refractory due to IFN as $\sigma =1.97\times 10^{-6}$ per day.

### 3.3. Basic Reproduction Number

The basic reproduction number, $R_{0}$, represents the average number of infectious viruses generated by a single infectious virus when target cells are not limiting [57,58,59]. For Model 1, $R_{0}$ can be calculated using the formula [49]
(4)$$R_{0}=\frac{\alpha p\beta T_{0}}{c\delta }.$$

The basic reproduction number, $R_{0}$, for SARS-CoV-2 estimated by our modeling varies among individual ferrets from 1.52 to 2.69, with an average value of 2.05 (Table 4). For within-host virus dynamics, this information can be applied to determine whether a virus can establish infection [57,58,59]. In general, if $R_{0}<1,$ the infection will die out, and if $R_{0}>1$, the infection will spread [57,58,59]. Consistent with this theoretical result, our estimate also showed each $R_{0}$ to be greater than 1 as the infection was established in each ferret. In addition, $R_{0}$ estimate can also be applied to calculate the effectiveness of antiviral agents, such as a vaccine, required to stop the viral replication and avoid infection. For example, the average value of $R_{0}=2.05$ indicates that at least 51%, i.e., ($1-1/R_{0})$, efficacy is required for antiviral agents blocking viral entry or production to stop the infection.

### 3.4. Viral Kinetic Properties

We also used our model to estimate other kinetic properties, including pre-peak growth rate and doubling time as well as post-peak decay rate and half-life of infectious virus (Table 4), during the SARS-CoV-2 infection in ferrets. During the pre-peak phase, the concentration of infectious virus grew exponentially, with a per capita growth rate of 2.29 per day (Table 4), resulting in a doubling time of 7.27 h. Similarly, during the post-peak phase, the infectious virus decayed with a per capita decay rate of 1.95 per day (Table 4) and a half-life of 8.54 h. In addition to these rates, we also estimated the net target cell loss in the upper respiratory tract of ferrets due to SARS-CoV-2 infection (Table 4). On average, approximately 78% of target cells were lost in the upper respiratory tract during the entire infectious period of SARS-CoV-2 in ferrets.

### 3.5. Comparison between F13-E and CTan-H Viruses

We compared the viral dynamics generated due to infection by the two types of viruses, F13-E and CTan-H. As discussed above, these virus types represent viruses in the environment (seafood market of Wuhan) and viruses within an infected person, respectively. Successful establishment of infection by both viruses ($R_{0}>1$ in both cases) implies that environmental transmission may also be an important route in addition to direct transmission. We observed a similar trend of viral dynamics in both cases (Figure 1), and the estimates of the parameters did not show any statistically significant difference between the two virus infections (*t*-test, *p*-value ≥0.05). Furthermore, a comparison of the kinetic parameters in all three models (Table 1, Table 2 and Table 3) revealed that none of the parameters were statistically different between two viruses (*t*-test, *p*-value ≥0.05). These results combined indicate that the environmental virus (F13-E) was as infectious as the within-human virus (CTan-H), causing similar cell loss and other properties of infection.

### 3.6. Sensitivity Analysis

Even though we estimated model parameters using experimental data, there is uncertainty among the estimated parameters, and the limited number of animals used in the experiment may not be representative of a larger population. Therefore, we performed a global sensitivity analysis of each model parameter using the Latin Hypercube sampling technique. Specifically, we computed partial rank correlation coefficients (PRCC) for each parameter of all three models corresponding to total viral-peak, infectious viral-peak, time to viral-peak, and infected target cell loss (Figure 4). In general, the infection rate (β), the virus production rate (p), the proportion of infectious virus (α), and the rate of transfer from eclipse phase to infectious phase (k) positively impacted total viral-peak, infectious viral-peak, and cell loss and negatively impacted time to viral-peak. The viral clearance rate (c), the infected cell death rate (δ), and the IFN-induced antiviral efficacy (σ) have both positive and negative effects depending on the property being examined (Figure 4). As observed from all the models, compared to the basic viral dynamic parameters (c, p, α, δ, β), the eclipse phase related parameter (k) and the immune-response-related parameter (σ) have significantly lower impact on the viral dynamics (Figure 4), supporting the results above that additional parameters in eclipse phase model (Model 2) and immune response model (Model 3) did not improve the data-fit compared to the basic viral dynamics model (Model 1).

### 3.7. Initial Infectious Virus

As mentioned earlier, animals were inoculated intranasally with ~$10^{5}$ PFU in a volume of 1 mL, but our estimates showed a relatively low level of initial infectious virus at the infection site (8.18 PFU/mL for F-13E group and 2.46 PFU/mL for CTan-H group). To understand this discrepancy, we performed further data fitting by fixing the initial infectious virus population at the infection site (upper respiratory tract) at $V_{i0}=10^{2},\;10^{3},10^{4}$ PFU/mL (Figure 5). We found that a low level of initial infectious virus is needed to obtain reasonable fits to the data (F-13E-2: SSR = 3.21, 3.99, 7.65, 10.65, AIC = 28.64, 30.80, 37.32, 40.63, for $V_{i0}=9.25,\;10^{2},\;10^{3},10^{4}$ PFU/mL, respectively; CTan-H-2: SSR = 2.99, 3.64, 5.79, 8.21, AIC = 27.92, 29.89, 34.53, 38.03, for $V_{i0}=9.12,\;10^{2},\;10^{3},10^{4}$ PFU/mL, respectively). There may be a number of factors that have contributed to this discrepancy. The viral stocks used for inoculation were prepared in Vero E6 cells and the measurement of their infectiousness was done by plaque assay using Vero E6 cells [1]. Vero E6 cells are a stable cell line derived from African green monkey kidney cells. The cells have lost the ability to secrete IFN due to spontaneous gene deletions [60,61], indicating Vero cells should be more susceptible to infection than the target cells in the ferret. Thus, PFUs measured in Vero cells may be considerably greater than the corresponding infectivity in ferrets. Additionally, the infectiousness at the infection site within ferrets may be different from the one estimated in in-vitro experiments, as the virus may adapt to the in-vivo cell environment [62]. Lastly, we note that the infection of SARS-CoV-2 has been successfully established in ferrets with widely varying initial infectious doses and the infection has been reported to depend on the route of transmission [63,64].

## 4. Discussion

Despite the development of vaccines against SARS-CoV-2, the threat of the COVID-19 pandemic remains. Thus, there is a continuing and urgent need to administer effective antiviral agents for SARS-CoV-2 treatment. Animal models are often used for experimental trials, while developing antiviral agents and ferrets are considered suitable hosts for respiratory diseases, including SARS-CoV-2. Here, we used mathematical models to characterize SARS-CoV-2 infection within ferrets infected with F13-E (SARS-CoV-2 isolated from an environmental sample) or infected with CTan-H (SARS-CoV-2 isolated from an infected individual). Our modeling results, including a global sensitivity analysis, have provided important insights into the viral dynamics and viral kinetics in ferrets and allowed us to compare them with infection in humans. Such information is useful for performing experiments in animals to design antiviral therapies for humans.

The temporal pattern of SARS-CoV-2 dynamics in ferrets is similar to the infection in humans [10,40,41,43,44] as well as to that of some other acute viral infections, such as influenza [36,49], though the infection in ferrets lasts for a significantly shorter duration than in humans (~10 days in ferrets [1] vs >20 days in humans [8,41,44]). Despite the short duration of infection, many of the viral dynamics parameters we estimated in ferrets, including infection rate, virus production rate, viral clearance rate, infected cell death rate, duration in eclipse phase, and IFN-induced antiviral efficacy, are similar to those in humans [8,40,44]. This similarity indicates that ferrets can be an appropriate animal model for experimental trials related to viral dynamics properties. It is worth noting that while we estimated the eclipse phase length and IFN-induced antiviral activities using extended models, our model comparison based on data fitting and the global sensitivity analysis showed that these additional parameters (impact of eclipse phase and IFN-induced antiviral activities) were not significantly impactful for infection dynamics in ferrets as compared to other basic viral dynamics parameters. Minimal effects of SARS-CoV-2-specific antibodies were also found in a previous study [9].

Because of the similarity in the basic viral dynamics parameters between SARS-CoV-2 in ferrets and humans and in other acute viral infections, such as influenza, the basic reproduction number and other viral kinetic properties (pre-peak growth rate, doubling time, post-peak decay rate, and half-life of infectious virus) are in a similar range [41,44]. However, the kinetic properties are quite different from other viral infections, such as HBV and EIAV, which result in chronic infection in the host [38,39]. For example, our estimate of the basic reproduction number in SARS-CoV-2 infection in ferrets ($R_{0}=2.05$) is lower than HBV infection in chimpanzee ($R_{0}>15$) [39] and EIAV infection in horses ($R_{0}>15$) [38]. Additionally, in EIAV [38], the infection cell death rate is negligible (~0.06 per day), resulting in only 0.7% cell loss. In contrast, SARS-CoV-2 infection causes significantly higher cell death (~5.20 per day) in ferrets, comparable to human influenza (~4 per day [49]), resulting in approximately 78% target cell loss in the upper respiratory tract of ferrets.

We also note that the initial decline in total virus observed in our simulations using the best-fit parameters is significantly larger compared to declines predicted by model fits to data with other viruses such as influenza. In the experiment modeled here, ferrets were infected with 10^5^ PFU in 1 mL of fluid [1]. Comparing the early measurements of the total virus with the measurements of infectious virus, we deduced that 1 PFU corresponds to approximately 10^4^ virions, which is in line with published estimates [56]; thus, we fixed $V_{0}$ at 10^9^ per mL. The first measured values were at day 2, and by then, viral loads were of order 10^6^ per mL, suggesting that there must be a dramatic decline in the viral load over the first two days, which our model replicates. The viral load decline very early in infection can be better estimated if more frequent sampling is done, especially within the first two days post-infection.

Our comparative analysis of viral dynamics initiated with a virus isolated from the environment and from a human host shows that both viruses can establish infection within the ferret, and the viral dynamics parameters in both cases are similar. This suggests the possibility of SARS-CoV-2 transmission from fomites. In both cases, we estimated that only about 0.02% of the newly produced virus was infectious, and the remaining portion was non-infectious. In addition, our estimate of $V_{i0}$ and our sensitivity analysis suggest that a small number of infectious SARS-CoV-2 at the infection site (upper respiratory tract) initiated infection despite a higher virus titer in the inoculum. These results indicate that ferrets need to be exposed to a large amount of virus for successful infections, consistent with previous studies, in which a small viral dose was not sufficient to infect all the inoculated animals in their experiment [63].

We acknowledge some limitations of our study. Our parameter estimates are based on a limited number of measurements from a small number of animals. Therefore, our estimates of parameters may have uncertainty that may arise from variation across animals. However, we performed a global sensitivity analysis using a wide parameter space, which suggests robustness of the viral dynamics and viral kinetic properties. While our fitting did not support the need to include an immune response in the model, the anti-SARS-CoV-2 antibody only measured for the viral level was undetectable in these ferrets, i.e., after 13 days or 20 days. More data with frequent measurements of immune responses can help improve our modeling and discern more accurately if an immune response is needed to clear this virus in ferrets. We also note that the parameter estimates we obtained may not reflect in viral kinetics in humans quantitatively because of extremely short-time dynamics in ferrets compared to humans. More focus on the qualitative conclusion of our study may be appropriate to predict human infection until these models are extensively tested in human infection with frequent pre-peak data. The extension of our modeling study by including potential antiviral therapy in ferrets can be a future avenue in this research direction.

We recognize that the viral strains used in the study we modeled are not the current variants of concern and that it would be valuable to model the viral kinetics of these strains. However, new variant viruses, such as Alpha (B.1.1.7) and Beta (B.1.351), are being studied [30,31], and when appropriate data is available, our models fit to the data may provide insights into their within-host kinetics.

In summary, we implemented three different viral dynamic models to analyze the data from SARS-CoV-2 infection in ferrets. Estimated parameters, predicted viral kinetic properties, and the global sensitivity analysis over a wide parameter space included in this study may provide useful information for the development of therapies utilizing ferrets as animal models for experimental trials.

## Figures and Tables

**Figure 1 viruses-13-01635-f001:**
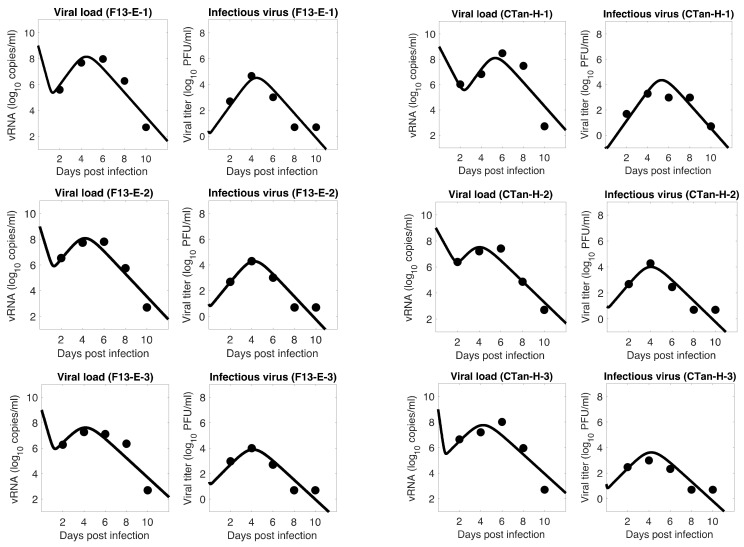
**Model 1 fitting to the data.** Prediction of the viral dynamics model 1 (solid line) along with the data (filled circle) containing the total viral load (RNA copies per mL) and the infectious virus titer (PFU per mL), collected from ferrets infected with F13-E SARS-CoV-2 and CTan-H SARS-CoV-2.

**Figure 2 viruses-13-01635-f002:**
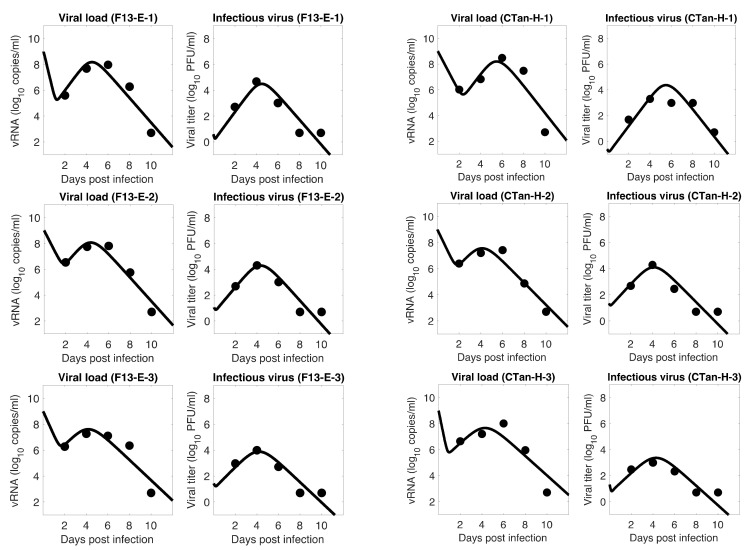
**Model 2 fitting to the data.** Prediction of the viral dynamics model 2 (solid line) along with the data (filled circle) containing the total viral load (RNA copies per mL) and the infectious virus titer (PFU per mL), collected from ferrets infected with F13-E SARS-CoV-2 and CTan-H SARS-CoV-2.

**Figure 3 viruses-13-01635-f003:**
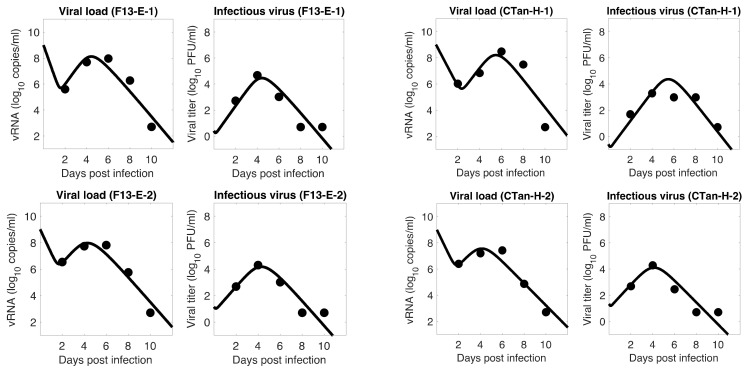
**Model 3 fitting to the data.** Prediction of the viral dynamics model 3 (solid line) along with the data (filled circle) containing the total viral load (RNA copies per mL) and the infectious virus titer (PFU per mL), collected from ferrets infected with F13-E SARS-CoV-2 and CTan-H SARS-CoV-2.

**Figure 4 viruses-13-01635-f004:**
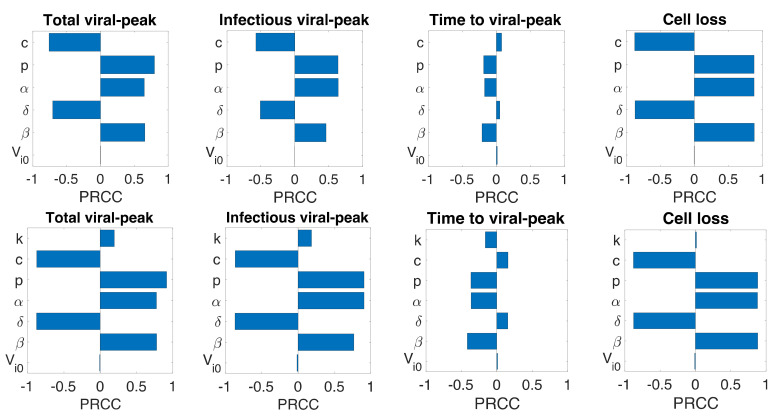
**Sensitivity analysis.** Partial rank correlation coefficients (PRCC) from the Latin hypercube sampling method of the total viral-peak size (**1st column**), infectious viral peak size (**2nd column**), time to viral-peak (**3rd column**), and cell loss (**4th column**) for Model 1 (**1st row**), Model 2 (**2nd Row**), and Model 3 (**3rd row**).

**Figure 5 viruses-13-01635-f005:**
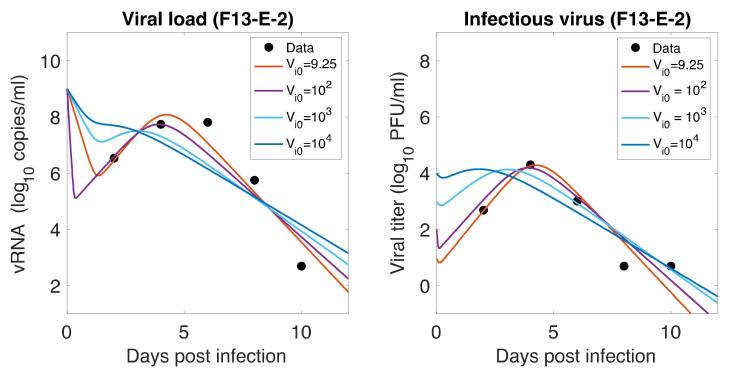
**Role of Initial infectious viruses.** Fitting data with initial virus at the infection site ($V_{i0}$) estimated vs fixed. Shown here are two representative animals from each of the two groups infected with F13-E SARS-CoV-2 and CTan-H SARS-CoV-2. The best-fit curves corresponding to $V_{i0}=9.25$ PFU/mL (F13-E-2) and $V_{i0}=9.12$ PFU/mL (CTan-H-2) were obtained by allowing $V_{i0}$ to be estimated by data fitting process, while other best-fit curves were obtained by fixing the values of $V_{i0}$ at $10^{2}$, $10^{3}$, and $10^{4}$ PFU/mL.

**Table 1 viruses-13-01635-t001:** Parameters of Model 1 estimated from the data fitting. G. Ave. stands for geometric average. *p*-values indicate the result of comparison between F13-E and CTan-H groups.

Ferret	$V_{i0}$ (PFU/mL)	β (PFU/mL)	δ (/Day)	α	p (/Cell/Day)	c (/Day)
F13-E SARS-CoV-2 virus						
F13-E-1	2.64	$3.55\times 10^{-5}$	4.05	$2.33\times 10^{-4}$	7934	7.13
	$\left(0.0002,\;51.75\right)$	$\left(1.16\times 10^{-5},\;9.99\times 10^{-5}\right)$	$\left(1.05,\;12.79\right)$	$\left(4.42\times 10^{-5},\;0.92\right)$	$\left(753,\;75,412\right)$	$\left(1.93,\;49.91\right)$
F13-E-2	9.25	$4.48\times 10^{-5}$	5.05	$1.63\times 10^{-4}$	8761	6.30
	$\left(0.12,\;116.55\right)$	$\left(1.66\times 10^{-5},\;9.98\times 10^{-5}\right)$	$\left(1.02,\;12.66\right)$	$\left(5.48\times 10^{-5},\;0.93\right)$	$\left(653,\;61,520\right)$	$\left(2.52,\;49.98\right)$
F13-E-3	22.39	$7.05\times 10^{-5}$	5.99	$1.90\times 10^{-4}$	5102	6.64
	$\left(0.0016,\;274.25\right)$	$\left(2.34\times 10^{-5},\;1.00\times 10^{-4}\right)$	$\left(1.11,\;11.56\right)$	$\left(4.84\times 10^{-5},\;0.91\right)$	$\left(765,\;54,913\right)$	$\left(3.28,\;49.98\right)$
G. Ave. (F13-E)	8.18	$4.82\times 10^{-5}$	4.97	$1.93\times 10^{-4}$	7078	6.68
CTan-H SARS-CoV-2 virus						
CTan-H-1	0.11	$6.50\times 10^{-5}$	4.89	$1.79\times 10^{-4}$	4152	3.67
	$\left(0.002,\;80.79\right)$	$\left(1.38\times 10^{-5},\;1.00\times 10^{-4}\right)$	$\left(0.91,\;19.99\right)$	$\left(2.38\times 10^{-5},\;0.89\right)$	$\left(632,\;94,492\right)$	$\left(2.22,\;49.96\right)$
CTan-H-2	9.12	$9.90\times 10^{-5}$	5.52	$3.06\times 10^{-4}$	1461	3.50
	$\left(0.05,\;231.2\right)$	$\left(2.26\times 10^{-5},\;1.00\times 10^{-4}\right)$	$\left(1.12,\;14.59\right)$	$\left(8.57\times 10^{-5},\;0.93\right)$	$\left(618,\;32,596\right)$	$\left(3.04,\;49.89\right)$
CTan-H-3	14.13	$9.99\times 10^{-5}$	5.99	$7.32\times 10^{-5}$	16,641	13.41
	$\left(0.48,\;736.69\right)$	$\left(3.37\times 10^{-5},\;1.00\times 10^{-4}\right)$	$\left(1.04,\;19.98\right)$	$\left(1.63\times 10^{-5},\;0.92\right)$	$\left(723,\;125,883\right)$	$\left(2.75,\;49.99\right)$
G. Ave. (CTan-H)	2.46	$8.63\times 10^{-5}$	5.45	$1.59\times 10^{-4}$	4656	5.57
Overall G. Ave.	4.48	$6.46\times 10^{-5}$	5.20	$1.75\times 10^{-4}$	5741	6.10
*p*-value	≥0.05	≥0.05	≥0.05	≥0.05	≥0.05	≥0.05

**Table 2 viruses-13-01635-t002:** Parameters of Model 2 estimated from the data fitting. G. Ave. stands for geometric average. *p*-values indicate the result of comparison between F13-E and CTan-H groups.

Ferret	$V_{i0}$ (PFU/mL)	β (PFU/mL)	δ (/Day)	α	p (/Cell/Day)	c (/Day)	k (/Day)
F13-E SARS-CoV-2 virus							
F13-E-1	3.94	$5.21\times 10^{-5}$	3.25	$2.07\times 10^{-4}$	7018	7.99	11.96
	$\left(1.36\times 10^{-5},\;32.58\right)$	$\left(1.83\times 10^{-5},\;2.67\times 10^{-4}\right)$	$\left(1.78,\;8.87\right)$	$\left(4.53\times 10^{-5},\;1.05\times 10^{-3}\right)$	$\left(4980,\;22,125\right)$	$\left(2.62,\;18.21\right)$	$\left(2.83,\;19.20\right)$
F13-E-2	11.09	$8.28\times 10^{-5}$	4.03	$1.65\times 10^{-4}$	3512	3.74	8.71
	$\left(2.54\times 10^{-6}\;,\;\;71.45\right)$	$\left(3.07\times 10^{-5},\;2.79\times 10^{-4}\right)$	$\left(1.77,\;9.92\right)$	$\left(5.44\times 10^{-5},\;6.19\times 10^{-4}\right)$	$\left(1353,\;15,579\right)$	$\left(2.12,\;13.94\right)$	$\left(2.68,\;19.98\right)$
F13-E-3	26.80	$1.40\times 10^{-4}$	3.67	$1.89\times 10^{-4}$	1545	4.28	10.99
	$\left(1.09\times 10^{-3},\;99.58\right)$	$\left(5.34\times 10^{-5},\;5.00\times 10^{-4}\right)$	$\left(1.47,\;9.95\right)$	$\left(4.55\times 10^{-5},\;7.79\times 10^{-4}\right)$	$\left(390,\;8088\right)$	$\left(1.83,\;21.05\right)$	$\left(2.26,\;19.99\right)$
G. Ave. (F13-E)	10.54	$8.45\times 10^{-5}$	3.64	$1.86\times 10^{-4}$	3365	5.04	10.46
CTan-H SARS-CoV-2 virus							
CTan-H-1	0.24	$1.06\times 10^{-4}$	4.98	$1.44\times 10^{-4}$	5091	3.65	3.93
	$\left(3.26\times 10^{-4},\;10.26\right)$	$\left(1.98\times 10^{-5},\;4.98\times 10^{-4}\right)$	$\left(2.13,\;9.98\right)$	$\left(3.17\times 10^{-5},\;1.38\times 10^{-3}\right)$	$\left(1946,\;39,011\right)$	$\left(2.38,\;14.84\right)$	$\left(1.72,\;19.52\right)$
CTan-H-2	21.76	$8.02\times 10^{-5}$	6.42	$3.50\times 10^{-4}$	2143	3.98	12.52
	$\left(3.48\times 10^{-3},\;99.97\right)$	$\left(1.63\times 10^{-6},\;4.87\times 10^{-4}\right)$	$\left(1.77,\;9.99\right)$	$\left(1.01\times 10^{-4},\;7.49\times 10^{-3}\right)$	$\left(273,\;6474\right)$	$\left(1.28,\;16.24\right)$	$\left(2.58,\;19.99\right)$
CTan-H-3	21.03	$3.19\times 10^{-4}$	4.41	$5.07\times 10^{-5}$	4971	9.38	4.25
	$\left(0.03,\;95.20\right)$	$\left(1.26\times 10^{-4},\;5.00\times 10^{-4}\right)$	$\left(1.54,\;9.89\right)$	$\left(1.82\times 10^{-5},\;1.97\times 10^{-4}\right)$	$\left(4962,\;5376\right)$	$\left(2.51,\;30.18\right)$	$\left(1.61,\;19.99\right)$
G. Ave. (CTan-H)	4.80	$1.49\times 10^{-4}$	5.20	$1.37\times 10^{-4}$	3785	5.15	5.93
Overall G. Ave.	7.11	$1.12\times 10^{-4}$	4.35	$1.60\times 10^{-4}$	3569	5.09	7.88
*p*-value	≥0.05	≥0.05	≥0.05	≥0.05	≥0.05	≥0.05	≥0.05

**Table 3 viruses-13-01635-t003:** Parameters of Model 3 estimated from the data fitting. G. Ave. stands for geometric average. *p*-values indicate the result of comparison between F13-E and CTan-H groups.

Ferret	$V_{i0}$ (PFU/mL)	β (PFU/mL)	δ (/Day)	α	p (/Cell/Day)	c (/Day)	σ (/IFN/Day)
F13-E SARS-CoV-2 virus							
F13-E-1	2.53	$4.74\times 10^{-5}$	2.42	$2.20\times 10^{-4}$	3980	5.43	$2.99\times 10^{-7}$
	$\left(6.13\times 10^{-4},\;35.91\right)$	$\left(1.07\times 10^{-5},\;1.88\times 10^{-4}\right)$	$\left(1.66,\;8.09\right)$	$\left(5.63\times 10^{-5},\;1.39\times 10^{-3}\right)$	$\left(3945,\;12,712\right)$	$\left(2.51,\;20.02\right)$	$\left(4.73\times 10^{-12},\;2.54\times 10^{-6}\right)$
F13-E-2	14.35	$4.32\times 10^{-5}$	2.22	$1.62\times 10^{-4}$	3984	4.12	$1.05\times 10^{-6}$
	$\left(8.38\times 10^{-3},\;\;92.33\right)$	$\left(1.51\times 10^{-5},\;2.11\times 10^{-4}\right)$	$\left(1.70,\;4.86\right)$	$\left(4.77\times 10^{-5},\;7.36\times 10^{-4}\right)$	$\left(2987,\;13,169\right)$	$\left(2.12,\;8.61\right)$	$\left(3.21\times 10^{-12},\;5.72\times 10^{-6}\right)$
F13-E-3	95.79	$3.79\times 10^{-5}$	2.26	$1.97\times 10^{-4}$	3951	6.25	$1.12\times 10^{-6}$
	$\left(14.37,\;96.68\right)$	$\left(9.60\times 10^{-6},\;1.36\times 10^{-4}\right)$	$\left(1.18,\;4.72\right)$	$\left(6.58\times 10^{-5},\;9.20\times 10^{-4}\right)$	$\left(3942,\;4063\right)$	$\left(2.23,\;29.06\right)$	$\left(8.35\times 10^{-8},\;8.29\times 10^{-6}\right)$
G. Ave. (F13-E)	15.16	$4.27\times 10^{-5}$	2.30	$1.92\times 10^{-4}$	3972	5.19	$7.06\times 10^{-7}$
CTan-H SARS-CoV-2 virus							
CTan-H-1	1.72	$1.55\times 10^{-5}$	4.62	$1.76\times 10^{-4}$	14,140	4.07	$3.38\times 10^{-6}$
	$\left(0.01,\;11.12\right)$	$\left(5.55\times 10^{-6},\;9.04\times 10^{-5}\right)$	$\left(2.06,\;9.96\right)$	$\left(3.04\times 10^{-5},\;6.92\times 10^{-4}\right)$	$\left(13,996,\;16,093\right)$	$\left(2.68,\;11.78\right)$	$\left(1.18\times 10^{-7},\;9.98\times 10^{-6}\right)$
CTan-H-2	16.22	$1.90\times 10^{-5}$	1.86	$3.18\times 10^{-4}$	3983	3.72	$6.72\times 10^{-6}$
	$\left(4.83\times 10^{-3},\;97.54\right)$	$\left(6.85\times 10^{-6},\;1.89\times 10^{-4}\;\right)$	$\left(1.48,\;5.96\right)$	$\left(8.02\times 10^{-5},\;1.22\times 10^{-3}\right)$	$\left(2955,\;4070\right)$	$\left(2.15,\;23.20\right)$	$\left(6.40\times 10^{-11},\;1.00\times 10^{-5}\right)$
CTan-H-3	35.21	$2.37\times 10^{-5}$	2.96	$6.03\times 10^{-5}$	14,100	3.48	$7.30\times 10^{-6}$
	$\left(0.23,\;99.99\right)$	$\left(9.95\times 10^{-6},\;2.81\times 10^{-4}\right)$	$\left(1.41,\;9.92\right)$	$\left(1.52\times 10^{-5},\;\;2.95\times 10^{-4}\right)$	$\left(14,066,\;14,121\right)$	$\left(2.00,\;30.67\right)$	$\left(2.07\times 10^{-7},\;1.00\times 10^{-5}\right)$
G. Ave. (CTan-H)	9.95	$1.91\times 10^{-5}$	2.94	$1.50\times 10^{-4}$	9270	3.75	$5.50\times 10^{-6}$
Overall G. Ave.	12.28	$2.85\times 10^{-5}$	2.60	$1.70\times 10^{-4}$	6068	4.41	$1.97\times 10^{-6}$
*p*-value	≥0.05	≥0.05	≥0.05	≥0.05	≥0.05	≥0.05	≥0.05

**Table 4 viruses-13-01635-t004:** Basic reproduction number ($R_{0}$), pre-peak infectious viral growth rate ($r_{g}$ ), post-peak infectious viral decay rate ($r_{d}$ ), pre-peak infectious viral doubling time ($t_{2}$ ), and post-peak infectious virus half-life ($t_{1/2}$ ) for each ferret. The growth rate and decay rates are computed using the eigenvalues of the Jacobian: $r\;=\;\left(-\left(c+\delta \right)\;+\;\sqrt{{\left(c+\delta \right)}^{2}\;+\;4c\delta \left(R_{t}-1\right)}\right)/2,\;$ where $R_{t}$ is approximation to the pre-peak and post-peak reproduction numbers. G. Ave. stands for geometric average. *p*-values indicate the result of comparison between F13-E and CTan-H groups.

Ferret	$R_{0}$	$r_{g}$	$r_{d}$	$t_{2}$	$t_{1/2}$	Cell Loss (%)
F13-E SARS-CoV-2 virus						
F13-E-1	2.27	2.66	2.17	6.26	7.68	85.76
F13-E-2	2.01	2.34	2.04	7.10	8.14	79.98
F13-E-3	1.72	1.95	1.77	8.52	9.38	69.95
G. Ave. (F13)	1.99	2.30	1.99	7.24	8.37	78.28
CTan-H SARS-CoV-2 virus						
CTan-H-1	2.69	2.70	2.16	6.16	7.69	91.48
CTan-H-2	2.32	2.26	1.85	7.37	8.99	86.48
CTan-H-3	1.52	1.94	1.74	8.55	9.54	59.35
G. Ave. (CTan-H)	2.12	2.28	1.91	7.29	8.71	77.72
Overall G. Ave.	2.05	2.29	1.95	7.27	8.54	78.00
*p*-value	≥0.05	≥0.05	≥0.05	≥0.05	≥0.05	≥0.05

## Data Availability

The study did not report any data. Only digitized published data were used.

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
