# Peer review of "Modeling Within-Host Dynamics of SARS-CoV-2 Infection: A Case Study in Ferrets"

_viruses, 2021, doi:10.3390/v13081635_

Round 1

Reviewer 1 Report

Vaidya et al. developed mathematical models of within-host viral infection to estimate biological parameters driving the infection dynamics and infection-related characteristics for SARS-CoV-2 based on ferrets data. The emergence and spread of Covid-19 have caused significantly impacts on public health and economy, therefore accurate quantification of those biological parameters is important to better understand the virus and predict infection outcomes in various settings including drug treatment and vaccination. This manuscript aimed to contribute to this area. The manuscript is well written and the easy to follow.

However, there are several major issues to address for publication:

  1. Parameter identifiability: It’s known that the viral load data are insufficient to unambiguously determine all the parameters in the TIV model or its variates (such as the Model 2 and 3 in the manuscript) due to parameter correlations. For example, beta and p in the models are highly correlated such that neither of them can be accurately estimated by fitting to the viral load data unless one of them is known. The authors should address all non-identifiable issues before reporting the estimates of the parameters. Otherwise, any interpretation and discussion about the parameter estimates will be unreliable and sensitive to measurement errors.
  2. Model fits: Some model fits in Figure 1-3 exhibit a large decline during the early phase of infection. Although a similar transient decline was often observed in fitted TIV models for other types of virus (which is due to the death of the virions before their replications become dominant), such a decline normally took a much smaller time compared to the period of the exponential growth phase. I am concerned about the robustness of such a long decline phase and its plausibility from biological point of view.
  3. Method for model simulation and fitting: It is not clear if the authors conducted a proper searching method to find the global minimum. In addition, the authors mentioned RK4 was used for model simulation. I am wondering if they used the built-in ode45 solver (which is not using RK4) or coded RK4 for their purpose. I raise this issue because the TIV model may become a stiff problem for some regions of the parameter space and this can be problematic for parameter searching in model fitting. Therefore, I think the authors should use some solvers that also handle the stiff problem (such as the built-in ode15s, which will be much faster than RK4 for solving the TIV model) to redo the fitting and check if the model fits are robust to the numerical methods.

Author Response

We would like to thank the reviewers for their valuable comments and constructive suggestions, which have helped improve our manuscript significantly. Point-by-point responses to the reviewers' comments are provided below. The changes made in the revised manuscript are highlighted in red.

Response to Reviewer 1

General Comment: Vaidya et al. developed mathematical models of within-host viral infection to estimate biological parameters driving the infection dynamics and infection-related characteristics for SARS-CoV-2 based on ferrets' data. The emergence and spread of Covid-19 have caused significantly impacts on public health and economy, therefore accurate quantification of those biological parameters is important to better understand the virus and predict infection outcomes in various settings including drug treatment and vaccination. This manuscript aimed to contribute to this area. The manuscript is well written and the easy to follow.

Response: We thank the reviewer for careful reading of our manuscript and providing us with constructive suggestions. We have addressed all the comments provided by the reviewer. 

Comment-1: Parameter identifiability: It’s known that the viral load data are insufficient to unambiguously determine all the parameters in the TIV model or its variates (such as the Model 2 and 3 in the manuscript) due to parameter correlations. For example, beta and p in the models are highly correlated such that neither of them can be accurately estimated by fitting to the viral load data unless one of them is known. The authors should address all non-identifiable issues before reporting the estimates of the parameters. Otherwise, any interpretation and discussion about the parameter estimates will be unreliable and sensitive to measurement errors.

Response: We thank the reviewer for raising this important issue. While p and beta may be correlated for the TIV model, our model is a generalization that includes infectious virus and we fit to both the total VL as in the TIV model as well as to the measured infectious titer. Thus, we use more data than in the TIV model. To address the identifiability issue, we have performed a practical identifiability analysis and can show that both beta and p used to generate simulated data can be recovered by our fitting procedure. To quantify the correlation of parameters estimated using our model and available data sets (viral load and infectious virus), we also computed Pearson correlation coefficients for each model and for each animal using the parameters obtained from replicates by bootstrapping. Our results show that the parameters estimated in our case are not correlated. In the revised manuscript, we have provided correlation coefficients in Supplementary Information and discussed the identifiable issue in the text. (page – 4, 5)               

Comment-2: Model fits: Some model fits in Figure 1-3 exhibit a large decline during the early phase of infection. Although a similar transient decline was often observed in fitted TIV models for other types of virus (which is due to the death of the virions before their replications become dominant), such a decline normally took a much smaller time compared to the period of the exponential growth phase. I am concerned about the robustness of such a long decline phase and its plausibility from biological point of view.

Response: We agree with the reviewer that the large initial decline in total virus is indeed unusual compared to fits with other viruses. The authors of the original experimental paper state that they infected the animals with  PFU in one ml of fluid. Comparing the early measurements of total virus with the measurements of infectious virus we deduced that 1 PFU corresponds to approximately 104 virions, which is in line with published estimates [see Sender et al. PNAS 118, e2024815118 (2021)], thus we fixed V0 at 109 per ml. The first measured values were at day 2 and by then VLs were of order 106 per ml suggesting that there must be a dramatic decline in the VL over the first 2 days, which our model replicates. It would have been nice to have more frequent data, especially within the first 2 days, but that is not available. If there were a smaller initial decline as the reviewer seems to think there should be then the VL on day 2 would be higher than that given by the data. We added a comment in the discussion about this large initial decline in VL. (page -14)

Comment-3: Method for model simulation and fitting: It is not clear if the authors conducted a proper searching method to find the global minimum. In addition, the authors mentioned RK4 was used for model simulation. I am wondering if they used the built-in ode45 solver (which is not using RK4) or coded RK4 for their purpose. I raise this issue because the TIV model may become a stiff problem for some regions of the parameter space and this can be problematic for parameter searching in model fitting. Therefore, I think the authors should use some solvers that also handle the stiff problem (such as the built-in ode15s, which will be much faster than RK4 for solving the TIV model) to redo the fitting and check if the model fits are robust to the numerical methods.

Response: We thank the reviewer for making us aware on the possibility of stiffness of the equations. We used the built-in ode45 solver in MATLAB. The ‘ode45’ is based on an explicit Runge-Kutta (4,5) formula, the Dormand-Prince pair. We also performed a global minimum search by providing various initial guesses distributed uniformly across the reasonable parameter limits. As suggested by the reviewer, we now also performed model solving and data fitting using ode15s, which can solve stiff equations, and we did not find any difference in our fits. We discuss this issue in the revised manuscript. (page – 4, 5)

Reviewer 2 Report

The paper deals with the forecasting of evolving of the SARS-CoV-2 infection and the course of the COVID-19 decease on the basis of a few well-known dynamic models represented by simple systems of the ODEs. The authors try to solve the appropriate inverse problems by the classic `palette' method. Then, they show that the proposed theoretical solutions fit the observed data with moderately good accuracy.

My main concern is as follows:

Although the authors consider quite a simple problem, it remains to be inverse. In this context seems to be strange, why they did not exploit some of the highly developed regularization techniques stemming from the fundamental works by Tikhonov or Lavrentyiev. I believe that augmenting the SSR criteria with some regularizers can help to increase the overall accuracy significantly.

In addition, I recommend comparing the proposed approach with recent results obtained with the employment of modern neural networks (perhaps, taking into account some recent papers published in Viruses).

Nevertheless, the topic seems to be highly relevant to the Viruses scope, and the reported results are interesting. Therefore, I think, the paper can be published provided the authors will agree to make the proposed revision.

Author Response

Note: This review report had not been uploaded by the time we received an email about review reports on July 13, 2021. So, we were not aware of this review report until the deadline for revision (10 days). We could see this additional report when we were uploading the revision based on the other two reviewers.  

General Comment: The paper deals with the forecasting of evolving of the SARS-CoV-2 infection and the course of the COVID-19 decease on the basis of a few well-known dynamic models represented by simple systems of the ODEs. The authors try to solve the appropriate inverse problems by the classic `palette' method. Then, they show that the proposed theoretical solutions fit the observed data with moderately good accuracy.

ResponseWe thank the reviewer for reviewing our manuscript.

Comment-1: Although the authors consider quite a simple problem, it remains to be inverse. In this context seems to be strange, why they did not exploit some of the highly developed regularization techniques stemming from the fundamental works by Tikhonov or Lavrentyiev. I believe that augmenting the SSR criteria with some regularizers can help to increase the overall accuracy significantly.

ResponseWe appreciate the reviewer suggesting an alternate approach (Regularization techniques) to explore. As the reviewer identified, the models we are using are simple problems. For such problems, the data fitting procedure we used has been widely applied for many viral dynamics models of various viruses. The success of these approaches is well established in the vast available literature about viral dynamics modeling. In the revised manuscript, we have performed global minimization, identifiability analysis, correlation analysis based on bootstrapping sampling, and global sensitivity analysis. We believe thorough analyses performed in our manuscript can provide important information about SAR-CoV-2 viral dynamics. Given that all the parameters in our model are practically identifiable, we do not see how any improvement can be achieved with the method by Tikhonov or Lavrentyiev, which were developed for large-scale ill-posed problems. Our problem is not ill-posed. Also, we are sure the reviewer knows these regularization techniques add bias to the parameter estimation procedure. While it is always better to explore many possible approaches, considering entirely different additional approaches for this data set is out of the scope of the current study, for which our focus is biological.

Comment-2: In addition, I recommend comparing the proposed approach with recent results obtained with the employment of modern neural networks (perhaps, taking into account some recent papers published in Viruses).

ResponseWe are not aware of any neural network approaches applied to the SARS-CoV-2 dynamics models in ferrets. We are not clear which references the reviewer is referring to.  However, we have compared our results with other studies, such as SARS-CoV-2 in humans, influenza virus, hepatitis B virus, and EIAV.

Comment-3: Nevertheless, the topic seems to be highly relevant to the Viruses scope, and the reported results are interesting. Therefore, I think, the paper can be published provided the authors will agree to make the proposed revision.

Response: We thank the reviewer for finding our results interesting, and the paper can be published with revision. We have significantly improved our approaches by performing the global minimization, identifiability analysis, correlation analysis based on bootstrapping sampling, and global sensitivity analysis.

Reviewer 3 Report

The authors have developed a mathematical model to describe characterizations of SARS-CoV-2 infection in ferrets and have inferred several interesting conclusions.  For example, which parameters make a difference (or do not make a difference) in overall outcome of virus production; and that perhaps only a very small fraction of the inoculum actually contributes to infection.

My main concern about the paper is the impact of the data and the findings.  It uses relatively old data and compares two very similar strains. I'm not sure how long it takes to generate and analyze the data from your model, but the impact of this study would be greatly enhanced if you could run your model using the circulating virus VOCs.  a couple of papers have already published variant data in ferrets (see below).  In addition, there are  multiple papers with variant challenges in hamsters and mice available.  It would also it would be interesting to look at infections of bats with a bat coronavirus as a comparison.

Alpha (B.1.1.7) variant: 4'-Fluorouridine is a broad-spectrum orally efficacious antiviral blocking respiratory syncytial virus and SARS-CoV-2 replication. doi:10.1101/2021.05.19.444875.

Beta (B.1.351) variant: Therapeutic effect of CT-P59 against SARS-CoV-2 South African variant. doi:10.1016/j.bbrc.2021.06.016

Some minor comments include:

Results 3.1

Line 163 - please state what kind of sample you are using to measure virus - I assume these are nasal washes?

Line 201 - please briefly explain in the text what SSR and AIC values are and why you calculate them.

Line 205 - please remind the reader in the text what kappa and omega represent

Results 3.3

Line 243 - please change to "varies among individual ferrets" 

Line 261 - please define upper respiratory tract - do you mean nasal epithelium?  data has shown that one can detect low levels of viral in the trachea but it does not produce measurable infectious virus.

3.7

Line 321 - It is misleading and inaccurate to say the viruses are less infectious in ferrets than in Vero cells. I think what you might be trying to say is that Vero cells are more susceptible to infection than the ferret turbinate.  It's the host response that is different, not the properties of the virus. however, it is an interesting explanation because we do not currently know how much virus is required to cause disease in people (or animals). However, if you use the lower dose of 10^3 pfu (as in your reference #63), I presume your estimate of initial infectious virus would be negative... how would you reconcile this?

Author Response

We would like to thank the reviewers for their valuable comments and constructive suggestions, which have helped improve our manuscript significantly. Point-by-point responses to the reviewers’ comments are provided below. The changes made in the revised manuscript are highlighted in red.

Response to the Reviewer

General Comment: The authors have developed a mathematical model to describe characterizations of SARS-CoV-2 infection in ferrets and have inferred several interesting conclusions. For example, which parameters make a difference (or do not make a difference) in overall outcome of virus production; and that perhaps only a very small fraction of the inoculum actually contributes to infection.

Response: We thank the reviewer for careful reading of our manuscript and providing us with constructive suggestions. We have addressed all the comments provided by the reviewer.

Comment-1: My main concern about the paper is the impact of the data and the findings. It uses relatively old data and compares two very similar strains. I'm not sure how long it takes to generate and analyze the data from your model, but the impact of this study would be greatly enhanced if you could run your model using the circulating virus VOCs. a couple of papers have already published variant data in ferrets (see below). In addition, there are multiple papers with variant challenges in hamsters and mice available. It would also be interesting to look at infections of bats with a bat coronavirus as a comparison.

Alpha (B.1.1.7) variant: 4'-Fluorouridine is a broadspectrum orally efficacious antiviral blocking respiratory syncytial virus and SARS-CoV-2 replication. doi:10.1101/2021.05.19.444875.

Beta (B.1.351) variant: Therapeutic effect of CT-P59 against SARS-CoV-2 South African variant. doi:10.1016/j.bbrc.2021.06.016

Response: We thank the reviewer for pointing this out and providing important references about new viral strains. We agree it would be valuable to model the viral kinetics with various VOCs. However, we do not have access to such data in ferrets and do not know of any that is publicly available. The two papers that the reviewer mentioned (the infection in ferrets and mice) are under therapy. The data collected in those papers are until day 4 only and also many of the VLs are undetected due to therapy. To use our models for those data, we require extra parameters (efficacy of therapy) as the efficacy was found to vary among different strains. Therefore, the data collected in those papers are insufficient, different and not comparable for our modeling, and the infection under therapy is not the focus of our paper. Hopefully, other modelers and experimentalists will examine such systems. However, the similar trends on viral dynamics with new VOCs observed in those papers (the reviewer mentioned) indicate that our parameter estimates may still provide important information for newly circulating virus strains. We have discussed these issues in the revised manuscript and cited the references provided by the reviewer. (page - 15)

Comment-2: (Results 3.1) Line 163 - please state what kind of sample you are using to measure virus - I assume these are nasal washes?

Response: It is nasal washes. We have provided this information in section “Experimental Data”. (page – 2).

Comment-3: Line 201 - please briefly explain in the text what SSR and AIC values are and why you calculate them.

Response: As suggested, we have restated in the results (Section 3.1) the meaning of SSR and AIC and their importance for model comparison purposes. (page – 9)

Comment-4: Line 205 - please remind the reader in the text what kappa and omega represent.

Response: We added the definition in the text. We think the reviewer meant sigma, not the omega in the text. (page – 10)

Comment-5: Results 3.3 Line 243 - please change to "varies among individual ferrets"

Response: We changed the sentence as suggested. (page – 10)

Comment-6: Line 261 - please define upper respiratory tract - do you mean nasal epithelium? data has shown that one can detect low levels of viral in the trachea but it does not produce measurable infectious virus.

Response: In the Shi et al. paper where our data comes from, virus was found in the nasal turbinate of one ferret and in the nasal turbinate, soft palate, tonsil and trachea of the other ferret that was euthanized.  The authors then state “These results indicates that SARS-CoV-2 can replicate in the uper respiratory tract of ferrets for up to 8 days …”. Thus, we use the term respiratory tract in the usual way. (page – 2)

Comment-7: Line 321 - It is misleading and inaccurate to say the viruses are less infectious in ferrets than in Vero cells. I think what you might be trying to say is that Vero cells are more susceptible to infection than the ferret turbinate. It's the host response that is different, not the properties of the virus. however, it is an interesting explanation because we do not currently know how much virus is required to cause disease in people (or animals). However, if you use the lower dose of 10^3 pfu (as in your reference #63), I presume your estimate of initial infectious virus would be negative... how would you reconcile this?

Response: We thank the reviewer for pointing this out. Yes, we wanted to say that Vero cells are more susceptible to infection than the target cells in the ferret. We have now changed the sentence to “The cells have lost the ability to secrete IFN due to spontaneous gene deletions [61,62], indicating Vero cells to be more susceptible to infection than the target cells in the ferret.Thus, PFUs measured in Vero cells may be considerably greater than the corresponding infectivity in ferrets.” (page – 12)

Our estimate of initial infectious virus is not due to the dose used, it is a result of the data used for fitting (minimization process). Our estimate is the best initial value that closely recovers these data. Forcing to have different initial infectious virus did not give the best fit to the model (see Figure 5). For this experiment, the dose of 10^5 pfu was used, thus we have to use this value to be consistent with the experiment. Even if we forcefully use lower dose of 10^3 pfu, we will not get negative initial infectious virus because the negative initial infectious virus does not generate the curves that are well fitted to the data. The minimization process will exclude that possibility. If we have enough viral load and infectious virus data points from the experiments initiated with lower dose, we can use our model to easily estimate how much infectious virus initiated the infection.

Round 2

Reviewer 1 Report

The authors have addressed all my comments.

Reviewer 3 Report

no additional suggestions.